# Research on Energy Supply Optimization of Diets for Songliao Black Growing and Fattening Pigs at a Low Ambient Temperature

**DOI:** 10.3390/ani15060846

**Published:** 2025-03-15

**Authors:** Zhaoyang Qi, Yu Zhang, Rui Han, Guixin Qin, Hailong Jiang, Dan Jiang, Dongsheng Che

**Affiliations:** Ministry of Education Laboratory of Animal Production and Quality Security, Jilin Provincial Key Laboratory of Animal Nutrition and Feed Science, College of Animal Science and Technology, Jilin Agricultural University, Changchun 130118, China; 18943117305@163.com (Z.Q.); 18943098852@163.com (Y.Z.); qgx@jlau.edu.cn (G.Q.); hljiang2024@163.com (H.J.); jiangdan126@163.com (D.J.)

**Keywords:** energy supply optimization, energy level, Songliao Black pig

## Abstract

At a low ambient temperature, appropriately increasing the fat level in the diet to enhance energy levels can affect the production performance and energy metabolism of pigs. To validate this hypothesis, we conducted a comprehensive study on Songliao Black growing and fattening pigs, measuring various parameters such as growth performance, slaughter performance, meat quality, nutrient digestibility, nitrogen balance, and energy metabolism. By analyzing these parameters, we aim to establish a robust foundation for optimizing the dietary nutrition plan for Songliao Black pigs in cold environments.

## 1. Introduction

The Songliao Black pig, as the first locally developed lean-meat-type black maternal line breed in northern China, is characterized by excellent reproductive performance, tender and flavorful meat, and remarkable adaptability to the harsh cold climates of northern regions. This indigenous breed currently represents the optimal genetic composition for meeting the demands of premium pork markets [1]. Due to the geographical location of northern China, the annual average temperature is only 7.6 °C, which is far below the lower limit of the optimal temperature for pigs (18 °C) [2]. In this case, in order for pigs to maintain normal production performance, the first solution is to consume mineral energy or electrical energy to maintain the normal temperature of the pigsty; the second is to increase dietary intake to compensate for the energy consumed to maintain normal body temperature. However, either method will consume additional resources and have adverse effects on the environment. Therefore, optimizing the dietary nutrition plan for pigs is an important technological approach to reduce production costs and greenhouse gas emissions in cold areas. Research has revealed significant differences between northern indigenous pig breeds and introduced exotic pig breeds in terms of the quantity and structure of energy and nutrient requirements, the metabolic utilization of energy and energy substrates, the characteristics of product quality formation, as well as the forms and quantities of pollutant emissions under low-temperature conditions [3,4,5]. Therefore, from the perspectives of resource utilization efficiency, production economic indicators, and environmental impacts, it is evident that Songliao Black pigs exhibit specific quantitative and structural demands for energy and nutrients under cold climatic conditions. Understanding the characteristics of energy demand of Songliao Black pigs under specific temperature conditions is of great significance for achieving precise nutrition plans for Songliao Black pigs.

At a low ambient temperature, the biological oxidation pathway is one of the main pathways for energy metabolism in animals, which oxidizes substrates mainly including carbohydrates, fats, and proteins. When the body uses ingested energy substances to generate heat, there is a certain sequence in the utilization of different energy carrier substances. In general, carbohydrates are preferentially utilized for oxidative energy supply. If glucose cannot meet the body’s energy needs, fat is needed for oxidative energy supply [6]. If there is enough fat oxidized in the body, fat oxidation provides energy at a priority level and plays a greater role in energy supply efficiency [7]. In addition, feeding high-fat diets to piglets in cold environments (8 °C) reduces carbon dioxide emissions by about 19 L per piglet per day compared to low-fat diets [8], and the heat consumption was smaller. When sugar and fat cannot meet the body’s needs, protein oxidation in the body will be used to provide energy, but using protein to produce heat will increase nitrogen emissions and increase economic costs. Therefore, in a low ambient temperature environment, it is crucial to increase energy levels by adding high fat to diets to improve energy utilization efficiency and reduce carbon dioxide and nitrogen emissions in pigs.

Previous studies had found that, at a low ambient temperature (10 °C), increasing the fat level and energy level could improve the energy utilization efficiency and reduce carbon dioxide emissions of Songliao Black pigs [9], but it was only a restricted feeding condition, and the amount of added fat was relatively high, resulting in higher costs. On this basis, we set a free feeding condition and the diet was optimized, and the study was proposed to explore the effect of increasing the fat content appropriately to improve the energy level on the production performance, nutrient digestibility, energy metabolism, and oxidation energy supply efficiency of various nutrients, carcass traits, and meat quality of Songliao Black growing and fattening pigs, while ensuring that the digestible protein and trace elements in the diet were the same. The goal was to provide a scientific basis for optimizing the dietary nutrition plan of Songliao Black pigs in cold environments.

## 2. Materials and Methods

### 2.1. Ethical Statement

The study was conducted according to the guidelines of the Declaration of Helsinki and approved by the Animal Welfare Ethics Committee of Jilin Agricultural University.

### 2.2. Experimental Design

Forty-eight 75-day-old Songliao Black growing barrows with an initial weight of 33.38 ± 1.29 kg were randomized into two groups, with four replicates in each group and six pigs in each replicate. Two groups (CON group: low fat, normal energy; TES group: high fat, high energy) were fed isonitrogenous diets with different energy levels and fat contents. The experimental animals were raised at the same ambient temperature (10 ± 1 °C) all day. After 5 days of pre-feeding, the formal experiment began. Four Songliao Black barrows weighing approximately 80 kg were selected from each group for a 5-day experimental period for digestibility and metabolism and respiratory calorimetry tests. All pigs were slaughtered simultaneously at the end of the 110-day experimental period (185 days of age) when their average body weight reached approximately 110 kg. During the growth stage (30–60 kg), the CON group contained a fat content of 3.15% with a digestible energy level of 14.20 MJ/kg; conversely, the TES group contained a fat content of 8.09% along with a digestible energy level of 15.34 MJ/kg, and the duration of this trial was 44 days. The diet was modified at the conclusion of the growth phase. In the fattening stage (60–110 kg), the CON group contained a fat content of 3.69%, resulting in a digestible energy level of 14.02 MJ/kg; meanwhile, the TES group maintained 8.33% fat alongside a digestible energy level of 15.14 MJ/kg, and the duration of this trial was 66 days.

### 2.3. Experimental Feed

The diets used in this experiment were the complete diets provided by the Songliao Black Pig Breeding Farm of the Animal Husbandry Branch of Jilin Academy of Agricultural Sciences in Changchun, China. According to nutrient requirements of swine in China [10], the diets were formulated to meet the nutritional requirements of growing and fattening pigs. The crude protein content in the diets was determined according to (GB/T 6432-2018) [11], the crude fat content in the diets was determined according to (GB/T 6433-2006) [12], the crude fiber content in the diets was determined according to (GB/T 6434-2006) [13], and the crude ash content in the diets is determined according to (GB/T 6438-2007) [14]. Additionally, the digestible energy level in the diets was calculated according to (GB/T 39235-2020) [15]. The detailed composition and nutritional levels of the diets during both growing and fattening stages are presented in Table 1.

### 2.4. Management of Experimental Animal Feeding

The growth test site was provided by the Animal Husbandry Branch of Jilin Academy of Agricultural Sciences in Changchun, China. The pigsty was maintained in a well-ventilated, dry, and clean condition. The temperature within the pigsty was regulated at 10 ± 1 °C, with relative humidity controlled at approximately 70%. One week prior to the commencement of the experiment, both the feeding troughs and flooring of the pigsty were thoroughly cleaned and disinfected, and all pigs received vaccinations. Subsequently, they were raised and managed according to standard practices employed on commercial pig farms. During the formal experimental period, two groups were established: one receiving conventional diets and another provided with high-fat and high-energy diets. During the trial period, daily feed intake was recorded using the weigh-back method. Pigs were provided ad libitum access to diets via an automated feeding system. To ensure continuous feed availability, the feeders were replenished twice daily (8:00 a.m. and 4:00 p.m.) with fresh diets; additionally, pigs had unrestricted access to water for drinking. At the end of the experiment, each pig was humanely euthanized for carcass trait assessment.

The digestion and metabolism and respiratory calorimetry tests were conducted in an open respiratory calorimeter at the Animal Husbandry Branch of Jilin Academy of Agricultural Sciences in Changchun, China. The indoor metabolic cage, material trough, and water tank were thoroughly cleaned and disinfected one week before the start of the experiment. Throughout the entire experiment, the indoor temperature was maintained at 10 ± 1 °C and the humidity was kept at around 70%. Feeding was scheduled at 8:00 a.m. every day, and pigs were free to eat and drink. Before feeding at 8:00 a.m. every day, all the previous day’s feces and urine were collected, and the amount of feces and urine excreted was recorded. The collected feces and urine were mixed every day, 10% of the total amount of feces and urine was taken out, nitrogen was fixed with 10% hydrochloric acid and 10% sulfuric acid, respectively, and then samples were stored in a −20 °C refrigerator. Finally, all the samples collected over 3 days were mixed. The mixed fecal sample needed to be dried in a 65 °C oven, crushed, sieved, and stored for subsequent indicator determination.

## 3. Sample Collection and Indicator Determination

### 3.1. Growth Performance

Following a 5-day pre-feeding period, the formal feeding experiment commenced. All pigs were weighed on an empty stomach both at the beginning and at the conclusion of the experiment. Daily health assessments of the pigs were conducted throughout the study. During this period, the amount of diets provided to each pigsty was meticulously recorded, and the remaining diets in each pigsty were weighed at 8:00 a.m. the following day. Daily dietary intake was recorded separately for both the growth and fattening phases. On the starting and ending days of these phases, all pigs were again weighed on an empty stomach, allowing for the calculation of average daily weight gain and feed-to-gain ratio.

Average daily weight gain = (final weight − initial weight) ÷ daysAverage daily feed intake = dietary intake during the formal trial period ÷ daysFeed-to-gain ratio = average daily dietary intake ÷ average daily weight gain

### 3.2. Nutrient Digestibility Rate

The protein content in feces and urine was determined according to the method specified in (GB/T 6432-2018) [11], while the crude fat content in feces was determined according to the method specified in (GB/T 6433-2006) [12]. The gross energy of the diets and the energy of feces were measured using an oxygen bomb calorimeter (Parr 6300 Calorimeter, Moline, IL, USA).Fat digestibility (%) = (fat intake − fat content in feces) ÷ fat intake × 100Protein digestibility (%) = (protein intake − protein content in feces) ÷ protein intake × 100Energy digestibility (%) = (gross energy intake of diets − energy content in feces) ÷ gross energy intake of diets × 100

### 3.3. Nitrogen Balance Test

In nitrogen balance tests, nitrogen emissions from gases were not considered because the amount of nitrogen exhaled and inhaled by the body is relatively stable. Fecal nitrogen and urinary nitrogen were measured using the Kjeltec 8400 Automatic Nitrogen Analyzer (FOSS, Hilleroed, Denmark) to calculate nitrogen balance under different feeding conditions. The calculation formula was as follows:Nitrogen deposition (g/d) = ingested nitrogen (g/d) − fecal nitrogen (g/d) − urinary nitrogen (g/d)Apparent nitrogen digestibility (%) = [ingested nitrogen (g/d) − fecal nitrogen (g/d)] ÷ ingested nitrogen (g/d) × 100Nitrogen deposition rate (%) = nitrogen deposition (g/d) ÷ ingested nitrogen (g/d) × 100protein deposition (g/d) = nitrogen deposition (g/d) × 6.25

### 3.4. Energy Metabolism Test

Based on the 24 h gas exchange data collected from each pig over three days, the respiratory quotient was the ratio between CO_2_ production and O_2_ consumption. According to Brouwer [16], the heat production (HP) could be calculated using the 24 h gas exchange values and urinary nitrogen (UN) content. The energy of urine was measured using an oxygen bomb calorimeter (Parr 6300 Calorimeter, Moline, IL, USA). The retained energy (RE) was the gross energy intake (GE) minus the energy lost in feces (FE), the energy lost in urine (UN), and the heat production (HP). The protein deposition energy was calculated based on the nitrogen deposition (g/d) from the nitrogen balance experiment multiplied by 6.25 and 23.86. The fat deposition energy was the difference between the total retained energy and the protein deposition energy. The calculation formulas were as follows:
O_2_ consumption (L/min) = [air entering the respiration chamber (L/min) × outdoor air O_2_% − Gas exiting the respiration chamber (L/min) × indoor O_2_%]CO_2_ production (L/min) = [air entering the respiration chamber (L/min) × indoor CO_2_% − Gas exiting the respiration chamber (L/min) × outdoor CO_2_%]Respiratory quotient (RQ) = CO_2_ production (L/d) ÷ O_2_ consumption (L/d)HP (kJ/d) = 16.18 × O_2_ (L/d) + 5.02 × CO_2_ (L/d) − 2.17 × CH4 (L/d) − 5.99 × UN (g/d)RE (kJ/d) = GE (kJ/d) − FE (kJ/d) − UN (kJ/d) − HP (kJ/d)Protein deposition energy (kJ/d) = nitrogen deposition (g/d) × 6.25 × 23.86 (kJ/g)Fat deposition energy (kJ/d) = RE (kJ/d) − protein deposition energy (kJ/d)

The oxidations of protein (OXPRO), carbohydrates (OXCHO), and fat (OXFAT) were calculated according to the method by Chwalibog et al. [17]. The calculation formulas were as follows:
Protein oxidation (OXPRO) (kJ) = UN (g) × 6.25 × 18.42 (kJ/g)Carbohydrate oxidation (OXCHO) (kJ) = [−2.968 × O_2_ (L) + 4.174 × CO_2_ (L) − 2.446 × UN (g)] × 17.58 (kJ/g)Fat oxidation (OXFAT) (kJ) = [(1.719 × O_2_ (L) − 1.719 × CO_2_ (L) − 1.963 × UN (g)] × 39.76 (kJ/g)

### 3.5. Collection and Measurement of Blood Samples

On the day before the end of the feeding trial, blood samples were collected from the jugular vein of pigs in the large pen using a syringe. The pigs were restrained with a restraint rope during this process, and all pigs were fasted. The collected blood was placed in 10 mL serum separator tubes and centrifuged at 3500× *g* for 15 min at 4 °C using a 5430 Eppendorf centrifuge (Hamburg, Germany). The supernatant was then aliquoted into 2 mL enzyme-free centrifuge tubes and stored in a −80 °C freezer.

The following parameters were measured using a fully automated biochemical analyzer (BS-400, Mindray, Shenzhen, China): alanine aminotransferase (ALT), aspartate aminotransferase (AST), total protein (TP), blood urea nitrogen (BUN), blood glucose (GLUC), high-density lipoprotein (HDL), low-density lipoprotein (LDL), and triglycerides (TG).

### 3.6. Slaughter Performance

On the day before the official trial period (110 d) ended, the fattening pigs were fasted for 12 h. On the last day, all fattening pigs were stunned using electric shock for humane euthanasia, followed by slaughter. The pre-slaughter live weight, carcass weight, dressing percentage, average backfat thickness, and eye muscle area were measured according to “Determination of Carcass Traits of Lean Pigs” (NY/T 825-2004) [18]. The live lean meat percentage and live fat percentage were calculated using the following formulas [19,20]:
Live lean meat mass (kg) = −6.9144 + 0.6154*X*_1_ − 2.6893*X*_2_, where *X*_1_ was the pre-slaughter live weight (kg); *X*_2_ was the backfat thickness between the 6th and 7th ribs (cm)Live fat mass (kg) = −26.4 + 0.221*EBW* + 1.331*P*_2_*,* where *EBW* was the empty live weight (kg); *EBW* = 0.905 × *BW*^1.013^; *BW* was the live weight (kg); *P*_2_ was the backfat thickness at the last rib of the left carcass (mm)Live lean meat percentage (%) = live lean meat mass ÷ pre-slaughter live weightLive fat percentage (%) = live fat mass ÷ pre-slaughter live weight

### 3.7. Meat Quality

The left half-carcass of pigs was suspended. An incision was made starting from the anterior end of the third thoracic vertebra (counting caudally) and extended posteriorly until the longissimus thoracis (LT) was isolated to meet the specified measurement criteria for length and weight. Meat quality parameters were determined as previously described [21,22,23]. The specific determination methods were as follows:

Meat color: Measurements were conducted within 45–60 min post mortem. Meat color was quantified using a chroma meter (45 min, Kinica Minolta Sensing Inc., Tokyo, Japan). The chroma meter was calibrated to the CIE Lab color system using the CR-A44 calibration plate (serial number 16433029), with a D65 light source, standard 2° observer, and an aperture of 8 mm. Two replicate slices per sample were analyzed, three distinct points were measured on each slice, and results were expressed as the arithmetic mean of six measurements.

Marbling score: Following the meat color measurement, the LT sample was placed on a porcelain tray to maintain surface integrity. The marbling score was determined using the guidelines of the National Pork Producers Council (NPPC). Each sample was divided into two replicate slices. A marbling score was independently assigned to each slice by trained evaluators. The final score was expressed as the mean of the two replicate assessments. Scoring should be completed within 30 min.

pH_24h_: The LT samples were stored at 0–4 °C and the pH value was measured at 24 h ± 10 min post mortem, denoted as pH_24h_. The meat pH was measured by a portable pH meter (TESTO205PH, TESTO Instruments, Lenzkirch, Germany) at 24 h. Two points were measured at each end of the specimen, and the average value was calculated.

Drip loss rate: For drip loss determination, the LT samples (about 55 g) were cut into 3.0 cm long cubes, weighed at 45 min post mortem, suspended in bags for 24 h at 4 °C, and reweighed. Four replicates were analyzed per sample, and the results were expressed as the arithmetic mean of the four measurements.

Cooking loss rate: From each LT sample, a 2.5 cm thick section was excised and trimmed into approximately 70 g cubes (5 cm × 5 cm × 2.5 cm) to ensure uniform cooking. The trimmed cubes were weighed, vacuum-sealed in plastic bags, and stored at 4 °C until subsequent analysis of cooking loss and Warner–Bratzler shear force (WBSF). Samples were first immersed in a 70 °C water bath for 35 min, followed by immediate cooling in an ice bath for 30 min. After cooling, samples were blotted dry with absorbent paper and reweighed to calculate cooking loss as a percentage of the pre-cooking weight. Four replicates were analyzed per sample, and the results were expressed as the arithmetic mean of the four measurements.

Shear force: Briefly, 150 g of LT samples was stored at 4 °C for 24 h, then randomly allocated to a constant-temperature water bath (80 °C) and cooked to an internal temperature of 70 °C. After cooling to room temperature, cooked samples were cut into ten cuboids (1 cm × 1 cm × 3 cm) parallel to the myofibril orientation. Shear force measurements were performed using a C-LT3B digital tenderness tester (Tenovo, Beijing, China) equipped with a Warner–Bratzler blade, a 15 kg load cell, and a crosshead speed of 200 mm/min. Three replicates were analyzed per sample, and the results were expressed as the arithmetic mean of the three measurements in newtons (N).

### 3.8. Statistical Analysis

The data obtained from this experiment were organized using Excel and analyzed using SPSS 25.0 (IBM-SPSS Inc., Chicago, IL, USA) for independent samples *t*-tests. The results were presented as mean ± standard deviation, with *p* < 0.05 indicating statistical significance.

## 4. Results

### 4.1. Growth Performance

From Table 2, it could be seen that the average daily feed intake of the TES group was lower compared to the CON group (*p* < 0.05). Additionally, the feed-to-gain ratio in the TES group was lower compared to the CON group (*p* < 0.01), with a decrease of 15.36%.

### 4.2. Nutrient Digestibility

From Table 3, it could be observed that the crude fat digestibility in the TES group was higher compared to the CON group (*p* < 0.01). Both energy digestibility and crude protein digestibility were improved to varying degrees, but they were not statistically different (*p* > 0.05).

### 4.3. Nitrogen Balance Test

From Table 4, it could be seen that the nitrogen intake in the TES group was lower compared to the CON group (*p* < 0.01). Additionally, the urine nitrogen content in the TES group was lower (*p* < 0.05). Although the apparent nitrogen digestibility and nitrogen deposition rate in the TES group showed varying degrees of improvement, and the fecal nitrogen content decreased, they were not different (*p* > 0.05).

### 4.4. Energy Metabolism

From Table 5, it could be observed that the urine energy content in the TES group was lower (*p* < 0.05), the deposition energy was higher (*p* < 0.01), the energy deposition rate was higher (*p* < 0.05), and the deposition energy of fat was higher (*p* < 0.05) compared to the CON group. In comparison to the CON group, the TES group showed a lower carbohydrate oxidation energy (*p* < 0.01) and protein oxidation energy (*p* < 0.05), while the fat oxidation energy was higher (*p* < 0.05). Additionally, the TES group had a daily carbon dioxide emission reduction of 44 L and a decrease in the RQ value, although they were not statistically different (*p* > 0.05). Furthermore, the TES group exhibited varying degrees of decrease in GE, HP, and FE compared to the CON group, but they were not statistically different (*p* > 0.05). The TES group also increased the deposition energy of protein, but this was not statistically different (*p* > 0.05).

### 4.5. Serum Biochemical Indicators

From Table 6, it could be observed that, compared to the CON group, the levels of blood urea nitrogen concentration in the TES group showed varying degrees of decrease, but there was no difference (*p* > 0.05). Additionally, the blood glucose concentration and low-density lipoprotein levels in the TES group exhibited an increasing trend, but they were also not different (*p* > 0.05). In comparison to the CON group, the TES group had a higher concentration of high-density lipoprotein and low-density lipoprotein (*p* < 0.05), as well as a lower concentration of alanine aminotransferase and aspartate aminotransferase (*p* < 0.05) and an extremely strong increase in triglyceride levels (*p* < 0.01). There was no difference in total protein concentration between the two groups (*p* > 0.05).

### 4.6. Slaughter Performance

From Table 7, it could be seen that, compared to the CON group, pre-slaughter live weight increased by 2.82% in the TES group, the carcass weight increased by 6%, and the dressing percentage improved by 2.26%, The live lean meat percentage showed a slight decrease, while the live fat percentage increased by 22.40%. The eye muscle area showed a slight increase, but it was not different (*p* > 0.05). In comparison to the CON group, both the live lean meat mass and live fat mass in the TES group exhibited an increasing trend, but they were not different (*p* > 0.05). However, the backfat thickness in the TES group was higher compared to the conventional dietary group (*p* < 0.05), with an increase of 20.90%.

### 4.7. Unit Body Composition and Consumption of Digestible Protein and Energy

To compare the effects of available nutrients per unit intake on weight gain, live lean meat mass, and live fat mass in pigs, we conducted a statistical analysis of the following indicators. According to Table 8, compared to the CON group, the weight gain per digestible protein was higher in the TES group (*p* < 0.01). Additionally, the live lean meat mass per digestible protein was higher than that in the TES group (*p* < 0.05). However, there were no differences between the two groups in terms of live fat mass per digestible protein, weight gain per digestible energy, live fat mass per digestible energy, and live lean meat mass per digestible energy (*p* > 0.05).

### 4.8. Meat Quality

From Table 9, it could be observed that, compared to the CON group, the yellowness (b*_45min_) value of the LT was higher in the TES group (*p* < 0.05), while the shear force of the LT was lower (*p* < 0.05). In comparison to the CON group, the brightness (L*_45min_) value, redness (a*_45min_) value, marbling score, and pH at 24 h in the TES group showed varying degrees of increase, but they were not different (*p* > 0.05). Additionally, the cooking loss rate and drip loss rate in the TES group were reduced to varying extents compared to the CON group, but again, they were not different (*p* > 0.05).

## 5. Discussion

When pigs are at a low ambient temperature, they will consume more energy to meet their metabolic requirements. It has been reported that the level of dietary fat had little effect on the production performance of growing and fattening pigs, but as the amount of fat added increased, the dietary intake showed a decreasing trend [24]. The dietary intake of the TES group during the growth and fattening periods in this study was lower than that of the CON group, which was consistent with the above study. This might be due to the TES group having high levels of fat, which provided more energy per kilogram of fat than carbohydrates [25]. This resulted in a reduction of dietary intake in the TES group. However, excessive fat supplementation may adversely affect dietary intake and health status in pigs, necessitating more trials to investigate the optimal dietary fat levels. The average daily weight gain of Songliao Black pigs in this study was not different during the growing and fattening periods. This might be because, under free feeding conditions, both groups consumed sufficient energy to meet their metabolic requirements. Han et al. [8] found that adding 5% fat to the diets of growing pigs at an ambient temperature of 13 °C did not increase daily weight gain, but could slightly reduce dietary intake and improve dietary utilization efficiency. In this study, the feed-to-gain ratio of pigs in the TES group was lower than that of the CON group, and the dietary utilization efficiency was higher, indicating that the added high fat could be better absorbed and utilized at a low ambient temperature. Moreover, it was found that the feed-to-gain ratio was more different during the fattening period than during the growth period, indicating that pigs had a more pronounced structural demand for energy and nutrition in the later stages of growth [26,27]. The added fat could be directly deposited into body fat, thereby eliminating the process of converting carbohydrates into fat and reducing energy loss. In summary, although the TES diet did not show benefits for the average daily weight gain of Songliao Black pigs, this benefit was reflected in the dietary utilization efficiency of the pigs.

Research has shown that, as the energy level of the diets increased, the crude fat digestibility of growing pigs tended to improve [28]. In addition, Chen et al. [29] found that, after treating the diet with a decreasing net energy gradient, the apparent digestibility of GE in the ileum in growing and fattening pigs decreased sequentially, and the apparent digestibility of crude protein in the ileum showed a decreasing trend. Kil et al. [24] found that, compared with the low-fat-supplemented control group, the high-fat-supplemented treatment group exhibited increased crude fat digestibility and GE digestibility in growing and finishing pigs. The crude fat digestibility of the TES group in this study was higher than that of the CON group, and both crude protein and energy digestibility were improved to varying degrees. This could be explained from two aspects: on the one hand, it might be because increasing dietary intake increased the passage rate of digestive substances in the gastrointestinal tract, reducing the time for diet to come into contact with digestive enzymes and the time for intestinal mucosa to absorb nutrients [30]; on the other hand, it might be due to endogenous loss leading to an increase in nutrients in feces [31], which explained the decrease in nutrient digestibility in pigs fed with the CON diet.

Researchers used 10%, 20%, and 30% rapeseed meal (11.1% fat) instead of soybean meal to feed piglets and found that, as the level of rapeseed meal increased, the content of fecal nitrogen increased linearly, while nitrogen digestibility decreased linearly. There were no differences in urine nitrogen, total nitrogen intake, and nitrogen deposition rate [32]. In this study, the urine nitrogen content of pigs in the TES group decreased, indicating a decrease in protein oxidation and more protein being absorbed and utilized. Due to the consistent protein levels in the two dietary groups, the TES group pigs had lower dietary intake than the CON group, resulting in a decrease in total nitrogen intake. In addition, most of the nitrogen deposited by fattening pigs in the later stage will be converted into protein in their muscles. The TES group pigs had a higher nitrogen deposition rate, which can verify the phenomenon of higher live lean meat mass in pigs. Through the analysis of the nutrient digestibility and nitrogen balance tests on Songliao Black pigs, it was found that, in a low-temperature environment, the TES diet could be better digested and utilized by the pigs, while also reducing the nitrogen content in urine and promoting nitrogen deposition.

The RQ value of pigs in the TES group was lower, indicating a higher proportion of fat oxidation energy supply. For every 1 g of glucose oxidized, 0.746 L of CO_2_ was produced, and for every 1 g of fat oxidized, 1.430 L of CO_2_ was produced [33], consuming 1 g of glucose and fat to supply 17.58 and 39.76 KJ of energy, respectively. When the same energy was released, the amount of CO_2_ produced by fat oxidation was less. The TES group received a high amount of fat, prioritized the oxidation of fat for energy supply in low-temperature environments, saved protein resources, and reduced carbon dioxide emissions, protecting the environment [9]. In addition, adding high fat increased the energy deposition rate, deposition energy, and deposition energy of fat of pigs, indicating that the energy intake of pigs in the TES group was better utilized, reducing energy loss and improving fat deposition efficiency. It was obvious that there was not much difference in deposition protein energy, but there was a difference in deposition fat energy. It was speculated that fat might replace some of the protein in the diets as a source of energy, thereby making protein more inclined to deposit in the muscles. However, when the energy supply level exceeded the production needs of pigs, the energy allocation pattern tended to deposit fat, thereby reducing live lean meat percentage [34]. The OXCHO and OXPRO of the TES group were lower than those of the CON group, while the oxidation level of OXFAT was higher than that of the CON group. This indicated that, at low ambient temperatures, although protein loss inevitably occurs due to metabolic turnover, the TES group was provided with sufficient dietary energy and fat was at a priority level in terms of energy supply ratio and had played a significant role in energy supply efficiency [7], thereby reducing the reliance on protein oxidation for energy. In contrast, the CON group experienced greater protein loss due to energy deficits and metabolic turnover, as evidenced by elevated urinary nitrogen excretion in the CON group. The increase in protein oxidation in the CON group could be demonstrated by the increase in urine nitrogen levels. Therefore, in a low-temperature environment, appropriately increasing the fat and energy levels in the diet could enhance the energy utilization efficiency and deposition efficiency of pigs, saving protein resources and reducing CO_2_ emissions.

Serum biochemical indicators are often used to evaluate the physiological condition, nutritional level, and health status of livestock [35]. The GLU, TP, and BUN are commonly used to evaluate energy metabolism and amino acid balance [36,37]. The concentrations of these two sets of indicators showed no statistical significance, indicating that there was no negative impact on energy and protein metabolism. The activity levels of AST and ALT can directly reflect whether animal liver function is impaired and whether the environment is comfortable [38]. The activity levels of AST and ALT in the CON group were slightly higher than those in the TES group in this study, which might be due to the fact that more protein in the CON group entered the liver and was oxidized under low-temperature conditions, leading to an increase in the activity of the two enzymes. Serum TG is closely related to energy metabolism in the body, reflecting the utilization of fat [39]. The serum TG content of the TES group in this study was higher than that of the CON group. This might be due to the increased intake of fats by pigs, which activated the fat transport mechanism in the pig’s body and led to an increase in serum TG levels. The HDL and LDL can reflect the lipid metabolism of the body. The HDL has the function of transporting cholesterol from surrounding tissues to the liver for breakdown, and an increase in LDL content increases the absorption and metabolism of adipose tissue, promoting animal growth [40]. The high-density lipoprotein and low-density lipoprotein in the TES group of this study were higher than those in the CON group. This indicated that pigs in the TES group had more deposition of fat and increased backfat thickness.

The slaughter performance is closely related to the fattening efficiency of growing and fattening pigs. Research had found that increasing NE levels by adding rapeseed oil increased the carcass weight and backfat thickness of fattening sows [41]. Some studies had also found that increasing the fat content in diets appropriately to improve energy levels had increased the backfat thickness and fat thickness of fattening pigs, and lean meat mass showed a downward trend [42]. Benz et al. found that, with the increase in fat content in the basal diet, the thickness of the 10th rib backfat increased, and the lean meat percentage showed a decreasing trend [43]. In addition, compared to the low-energy group with 0.5% fat addition, the high-energy group of fattening pigs with 4% fat addition had increased carcass weight and backfat thickness [44]. In this study, the TES group showed varying degrees of improvement in pre-slaughter live weight, carcass weight, slaughter rate, live fat percentage, and eye muscle area, increasing the backfat thickness of pigs. This could be verified by the magnitude of deposition energy, deposition energy of protein, and deposition energy of fat. It is noteworthy that the formulas used to estimate live lean meat mass and live fat mass have inherent limitations and may not be applicable to indigenous fat-type pig breeds. In summary, through the analysis of serum biochemical indicators and slaughter performance of Songliao Black pigs, it was found that the TES diet was beneficial for improving the fattening efficiency and backfat thickness of Songliao Black pigs, promoting deposition of fat.

From the analysis of the pork and protein deposition efficiency produced by each unit of effective nutrient intake, we found that, overall, the TES group outperformed the conventional dietary group in terms of weight gain/digestible protein, live fat mass/digestible protein, and live lean meat mass/digestible protein. This indicated that producing the same amount of pork consumed less protein, resulting in higher protein deposition efficiency, which saved protein resources and increased the deposition of lean meat and fat. However, the differences in the effects of the two groups of digestible energy on various production performance indicators were minor. This might be due to the fact that the energy deposition in fattening pigs was primarily reflected in fat deposition, which constituted a small proportion of carcass weight.

Pork quality is an important indicator for measuring the economic traits of growing and fattening pigs. Chen et al. [26] found that, compared with a low-energy diet supplemented with 3.2% soybean oil, a high-energy diet supplemented with 7.9% soybean oil increased the 24 h L* and b* values of pork and reduced cooking loss and drip loss of the LT. The b* value of the TES group in this study was higher than that of the CON group, which could be attributed to the high content of unsaturated fatty acids (UFAs) in soybean oil [45]. These UFAs were efficiently absorbed in the small intestine of pigs and subsequently deposited into body tissues as fatty acids [46]. The UFAs in the LT might be oxidized, leading to an increase in the b* value. The magnitude of shear force was used to evaluate the tenderness of meat. Research has found that shear force increased with decreasing dietary DE concentration [47,48]. The shear force of the TES group in this study was lower than that of the CON group, which was consistent with the aforementioned studies. In addition, Goh et al. [49] found that, as the pH value increased, both cooking loss and drip loss decreased. There was no difference in the drip loss rate between the two groups in this study, indicating that the two groups did not have any adverse effects on the water holding capacity of muscles. pH is an important indicator of muscle acidity, and a lower value indicates that more lactic acid is produced in the muscles, which is detrimental to meat quality. In this study, there was no difference in the pH values at 24 h between the two groups, and both were within the normal range (pH 5.3 to 6.8), indicating that neither group showed an adverse effect on the acidity of the meat quality. In short, the TES diet increased the tenderness of the LT in Songliao Black pigs.

## 6. Conclusions

In summary, improving the energy level by appropriately increasing the fat level in a low ambient temperature environment could enhance the production performance and energy utilization efficiency of Songliao Black pigs. This approach reduced CO_2_ emissions and protein oxidation, saved protein resources, and provided a scientific basis for a precise nutrition program of Songliao Black pigs.

## Figures and Tables

**Table 1 animals-15-00846-t001:** Basic diet composition and nutrition level (air-dried basis).

Items	Growing Stage (30–60 kg)		Fattening Stage (60–110 kg)	
	CON	TES	CON	TES
Corn	46.97%	37.59%	58.46%	47.47%
Corn starch	18.50%	19.90%	15.00%	17.97%
Wheat bran	7.35%	8.86%	6.10%	9.99%
Soybean meal	23.24%	24.44%	13.32%	13.89%
Calcium hydrogen phosphate	0.80%	0.80%	0.49%	0.46%
Stone flour	0.83%	0.82%	0.85%	0.93%
Salt	0.25%	0.25%	0.20%	0.20%
Soybean oil	0.70%	6.00%	1.00%	6.00%
Alfalfa meal	0.10%	0.10%	3.56%	1.89%
Lysine	0.39%	0.37%	0.31%	0.31%
Methionine	0.14%	0.15%	0.06%	0.08%
Threonine	0.14%	0.14%	0.10%	0.11%
Tryptophan	0.02%	0.02%	0.01%	0.02%
Valine	0.07%	0.06%	0.04%	0.18%
Premix ^(1)^	0.50%	0.50%	0.50%	0.50%
Total	100%	100.00%	100.00%	100.00%
Nutrient levels ^(2)^				
Digestible energy (MJ/kg)	14.20	15.34	14.02	15.14
Crude protein (%)	16.90	16.60	13.69	13.86
Ether extract (%)	3.15	8.09	3.69	8.33
Crude fiber (%)	5.59	4.45	3.73	3.38
Ash (%)	5.09	3.66	3.80	3.52
Digestible crude protein (%)	13.19	13.19	10.06	10.06
Lysine salt (%)	0.97	0.97	0.70	0.70
Tryptophan (%)	0.17	0.17	0.12	0.12
Methionine + cystine (%)	0.55	0.55	0.40	0.40
Threonine (%)	0.60	0.60	0.45	0.45
Calcium (%)	0.63	0.63	0.56	0.56
Available phosphorus (%)	0.27	0.27	0.19	0.19

^(1)^ The premix provided the following per kg of diet (30–60 kg): VD3 190 IU, VE 18 IU, VK 0.5 mg, choline 0.50 g, B6 1.5 mg, B12 15.0 μg, biotin 0.08 mg, folic acid 0.4 mg, nicotinic acid 15 mg, pantothenic acid 10 mg, thiamine 1.6 mg, riboflavin 3 mg, Cu 4.5 mg, I 0.14 mg, Fe 70 mg, Mn 3 mg, Se 0.30 mg, Zn 70 mg; The premix provided the following per kg of diet (60–110 kg): VA 1350 IU, VD3 160 IU, VE 14 IU, VK 0.5 mg, choline 0.40 g, B6 1 mg, B12 6.0 μg, biotin 0.07 mg, folic acid 0.3 mg, nicotinic acid 10 mg, pantothenic acid 8 mg, thiamine 1.5 mg, riboflavin 2 mg, Cu 3.5 mg, I 0.14 mg, Fe 50 mg, Mn 2 mg, Se 0.25 mg, Zn 50 mg; ^(2)^ CP, EE, CF, and ash were measured values, and the rest were calculated values.

**Table 2 animals-15-00846-t002:** Effect on growth performance of Songliao Black growing and fattening pigs.

Items	CON	TES	*p*-Value
30 to 60 kg Stage			
Initial Body Weight (kg)	34.06 ± 6.90	34.68 ± 4.21	0.745
Final Body Weight (kg)	58.70 ± 7.37	59.80 ± 5.84	0.624
Average Daily Weight Gain (kg/d)	0.57 ± 0.02	0.58 ± 0.03	0.388
Average Daily Feed Intake (kg/d)	2.07 ± 0.13 ^b^	1.89 ± 0.07 ^a^	0.02
Feed-to-Gain Ratio	3.63 ± 0.04	3.26 ± 0.03	0.08
60 to 110 kg Stage			
Initial Body Weight (kg)	59.23 ± 7.26	60.12 ± 6.40	0.679
Final Body Weight (kg)	107.28 ± 10.17	110.31 ± 8.93	0.316
Average Daily Weight Gain (kg/d)	0.74 ± 0.01	0.76 ± 0.02	0.163
Average Daily Feed Intake (kg/d)	2.93 ± 0.16 ^b^	2.63 ± 0.04 ^a^	0.01
Feed-to-Gain Ratio	3.98 ± 0.48 ^b^	3.45 ± 0.13 ^a^	<0.01

The absence of letters on the shoulder labels of peer data indicated no significant difference (*p* > 0.05), while different lowercase letters indicated significant differences (*p* < 0.05).

**Table 3 animals-15-00846-t003:** Effect on nutrient digestibility of Songliao Black fattening pigs.

Items	CON	TES	*p*-Value
Energy Digestibility (%)	86.49 ± 2.42	88.03 ± 1.75	0.422
Crude Protein Digestibility (%)	78.99 ± 3.08	80.88 ± 2.96	0.485
Crude Fat Digestibility (%)	67.16 ± 1.88 ^a^	84.38 ± 0.94 ^b^	<0.01

The absence of letters for the shoulder labels of peer data indicates no significant difference (*p* > 0.05), while different lowercase letters indicate significant differences (*p* < 0.05).

**Table 4 animals-15-00846-t004:** Effect on nitrogen balance of Songliao Black fattening pigs.

Items	CON	TES	*p*-Value
Intake Nitrogen (g/d)	61.53 ± 0.21 ^b^	59.51 ± 0.23 ^a^	<0.01
Fecal Nitrogen (g/d)	12.93 ± 1.94	11.38 ± 1.80	0.367
Urinary Nitrogen (g/d)	10.65 ± 1.08 ^b^	8.51 ± 1.02 ^a^	0.032
Apparent Nitrogen Digestibility (%)	78.99 ± 3.08	80.88 ± 2.96	0.485
Nitrogen Deposition Rate (%)	61.68 ± 2.30	66.59 ± 3.50	0.112

The absence of letters for the shoulder labels of peer data indicates no significant difference (*p* > 0.05), while different lowercase letters indicate significant differences (*p* < 0.05).

**Table 5 animals-15-00846-t005:** Effect on energy metabolism of Songliao Black fattening pigs.

Items	CON	TES	*p*-Value
Respiratory Quotient	0.98 ± 0.02	0.96 ± 0.02	0.358
Carbon Dioxide Emissions (L/d)	1065.96 ± 40.27	1021.86 ± 46.61	0.166
Oxygen Consumption (L/d)	1087.71 ± 28.60	1069.33 ± 49.02	0.792
Gross Energy (MJ/d)	49.75 ± 0.17	49.53 ± 0.19	0.206
Heat Production (MJ/d)	23.30 ± 0.63	22.03 ± 2.55	0.448
Fecal Energy (MJ/d)	6.72 ± 1.22	5.93 ± 0.89	0.415
Urinary Energy (MJ/d)	1.17 ± 0.03 ^b^	1.07 ± 0.02 ^a^	0.034
Deposition Energy (MJ/d)	18.55 ± 0.99 ^a^	21.94 ± 1.30 ^b^	0.003
Energy Deposition Rate (%)	37.30 ± 2.11 ^a^	42.43 ± 0.76 ^b^	0.017
Deposition Energy of Protein (MJ/d)	5.66 ± 0.19	5.91 ± 0.30	0.288
Deposition Energy of Fat (MJ/d)	12.89 ± 0.80 ^a^	15.93 ± 1.10 ^b^	0.018
OXCHO (MJ/d)	27.27 ± 1.90 ^b^	18.86 ± 2.24 ^a^	0.003
OXPRO (MJ/d)	1.17 ± 0.16 ^b^	0.88 ± 0.22 ^a^	0.032
OXFAT (MJ/d)	0.66 ± 0.03 ^a^	1.35 ± 0.02 ^b^	0.010

The absence of letters for the shoulder labels of peer data indicates no significant difference (*p* > 0.05), while different lowercase letters indicate significant differences (*p* < 0.05). OXCHO = carbohydrate oxidation; OXPRO = protein oxidation; OXFAT = fat oxidation.

**Table 6 animals-15-00846-t006:** Effect on serum biochemical indicators of Songliao Black fattening pigs.

Items	CON	TES	*p*-Value
ALT (U/L)	34.60 ± 1.52 ^b^	33.60 ± 2.70 ^a^	0.041
AST (U/L)	34.52 ± 1.21 ^b^	33.67 ± 2.53 ^a^	0.034
TP (g/L)	77.52 ± 4.75	77.77 ± 5.20	0.872
GLU (mmol/L)	4.60 ± 0.61	4.80 ± 0.83	0.386
BUN (mmol/L)	4.46 ± 0.48	4.23 ± 0.36	0.372
HDL (mmol/L)	0.98 ± 0.16 ^a^	1.10 ± 0.13 ^b^	0.012
LDL (mmol/L)	1.12 ± 0.15 ^a^	1.19 ± 0.15 ^b^	0.044
TG (mmol/L)	0.41 ± 0.16 ^a^	0.65 ± 0.23 ^b^	0.001

The absence of letters for the shoulder labels of peer data indicates no significant difference (*p* > 0.05), while different lowercase letters indicate significant differences (*p* < 0.05). ALT = alanine aminotransferase; AST = aspartate aminotransferase; TP = total protein; GLU = glucose; BUN = blood urea nitrogen; HDL = high-density lipoprotein; LDL = low-density lipoprotein; TG = triglyceride.

**Table 7 animals-15-00846-t007:** Effect on slaughter performance of Songliao Black fattening pigs.

Items	CON	TES	*p*-Value
Pre-slaughter Live Weight (kg)	107.28 ± 10.17	110.31 ± 8.93	0.316
Carcass Weight (kg)	76.32 ± 2.90	80.90 ± 2.43	0.053
Dressing Percentage (%)	71.72 ± 1.16	73.34 ± 1.02	0.132
Live Lean Meat Percentage (%)	50.65 ± 0.65	50.14 ± 0.54	0.266
Live Lean Meat Mass (kg)	54.09 ± 0.51	55.11 ± 0.67	0.079
Live Fat Percentage (%)	13.84 ± 3.13	16.94 ± 2.20	0.157
Live Fat Mass (kg)	14.13 ± 3.50	18.90 ± 2.43	0.084
Backfat Thickness (mm)	15.50 ± 1.96 ^a^	18.74 ± 0.99 ^b^	0.039
Eye Muscle Area (cm^2^)	40.12 ± 1.24	41.75 ± 1.97	0.210

The absence of letters for the shoulder labels of peer data indicates no significant difference (*p* > 0.05), while different lowercase letters indicate significant differences (*p* < 0.05).

**Table 8 animals-15-00846-t008:** Effects of available nutrients per unit intake on weight gain, live lean meat mass, and live fat mass in pigs.

Items	CON	TES	*p*-Value
Weight Gain/Digestible Protein (g/g)	17.40 ± 0.06 ^a^	19.80 ± 0.05 ^b^	0.006
Weight Gain/Digestibility Energy (g/MJ)	18.55 ± 0.60	19.48 ± 0.52	0.112
Live Fat Mass/Digestible Protein (g/g)	3.70 ± 0.09	5.00 ± 0.06	0.068
Live Fat Mass/Digestibility Energy (g/MJ)	4.00 ± 0.99	4.92 ± 0.63	0.166
Live Lean Meat Mass/Digestible Protein (g/g)	13.30 ± 0.08 ^a^	14.60 ± 0.02 ^b^	0.023
Live Lean Meat Mass/Digestibility Energy (g/MJ)	14.24 ± 0.84	14.36 ± 0.19	0.798

The absence of letters for the shoulder labels of peer data indicates no significant difference (*p* > 0.05), while different lowercase letters indicate significant differences (*p* < 0.05). The digestible energy and digestible protein in the table are calculated based on the feed formula table.

**Table 9 animals-15-00846-t009:** Effect on meat quality of Songliao Black fattening pigs.

Items	CON	TES	*p*-Value
L*_45min_	45.74 ± 2.71	46.07 ± 4.59	0.763
a*_45min_	6.33 ± 0.93	6.72 ± 0.89	0.185
b*_45min_	10.69 ± 0.69 ^a^	11.20 ± 0.74 ^b^	0.029
Marbling (score)	2.59 ± 0.36	2.66 ± 0.45	0.553
pH_24h_	5.51 ± 0.05	5.53 ± 0.04	0.231
Cooking Loss Rate (%)	28.35 ± 1.80	27.40 ± 2.51	0.202
Drip Loss Rate (%)	7.76 ± 2.28	7.70 ± 2.35	0.924
Shear Force (N)	49.86 ± 6.18 ^b^	45.66 ± 4.57 ^a^	0.023

The absence of letters for the shoulder labels of peer data indicates no significant difference (*p* > 0.05), while different lowercase letters indicate significant differences (*p* < 0.05). L*_45min_ = 45 min brightness value; a*_45min_ = 45 min redness value; b*_45min_ = 45 min yellowing value.

## Data Availability

The original contributions presented in the study are included in the article; further inquiries can be directed to the corresponding authors.

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
