# Peer review of "Research on Energy Supply Optimization of Diets for Songliao Black Growing and Fattening Pigs at a Low Ambient Temperature"

_animals, 2025, doi:10.3390/ani15060846_

Round 1
Reviewer 1 Report
Comments and Suggestions for Authors
Language should be revised for style and grammatical corrections by an language expert. Sentences in some parts ar somewhat difficult to follow. The main issue of the article is the use and the interpretation of the words »low ambient temperature«. Although the pigs were reared in low temperature environment, the onset of the trial is not such, that would allow this comparison (i.e. all animals were reared at the same temeprature). The authors should therefore reconsider and thoroughly rewrite the manuscript (especially discussion) to avoid any conclusions suggesting the effects of low temperature. The introduction is rather lenghty, unconnected and unfocused. It should be concentrated solely on providing main points to justify the aim of the study. On the other hand, it is lacking essential information theSongilao pig breed (main characteristics, main environment to which the breed is adapted…). Same goes for the discussio, authors should improve focus, avoiding lenghty descriptions that are not directly linked to the results. There are also other issues regarding the methods description, which are listed among specific comments below:
Title: you mean low ambient temperature environment?
Line 23: Please include the age.
Line 25: Please define the abbreviations CON and TES.
Line 31: were all animals slaughtered at the same time? What was their age and the duration of the trial?
Line 46: Please change the conclusions in line with the general comment.
Line 119: What was the age and sex of the pigs included.
Lines 122-131: Please include for all stages the time points or duraitons of the seperate phases . Were the changes in feed done at the same time points. Please also rewrite this part for better clarity.
Line 123: what was the sex of the animals selected (castrate, gilt?). It should always be the same, not to interfere with the metabolic results.
Line 128: It is better to adhere to the results of chemical composition (i.e. fat content) rather than confuse it with how much fat was added to the feed.
Line 164: Please describe how was the feed intake recorded, was the feeding on ad lib basis? How did you ensure the ad lib allowance if the feeding was done in the limited periods (i.e. twice a day)?
Line 249: From vhich vein exactly?
Line 251: What kind of tubes (anticoagulant) were used? If you are measuring glucose in blood, this additionally requires a glycolysis blocator.
Lines 255-256: What is the reason for applying these tests?
Line 259: what was the lenght of the trial?
Line 263: If the formula is for lean pigs, it does usually not cover the local breeds (fatty phenotypes). Please explain this (also in the discussion).
Lines 273-279: Please provide methodology more in detail (i.e. reference, basic principles, number of technical repetitions, sampling location, sampling time, with colour measurement time of exposure to air is also important).
Line 281: Please describe the statistical model and post-hoc tests.
Table 7: Effects on slaughter performance
Line 356 and Table 8 title: »composition consumes digestible protein…«? Rewrite.
Line 385: What is the real implication of this part? Please shorten considerably.
Line 395: By increasing the fat content, you also increase the energy content, which would block intake by itself in the long term. In addition dietary fats are the most potent regulator (blocator) of the apetite/feed intake. Please also consider that in the discussion, also in association to basic energy needs determined for the pigs from the experiment.
Line 426: You are refering to chicken. Please try finding relevant literature specifically for pigs.
Line 464: you are not showing any correlation in your results. Please rewrite.
Line 467: there are always some proteins degraded (obligatory protein loss) due to metabolism, turnover etc, even if there is enough energy available. The amount of protein that you are refering to would be replaced by fats if there was not enough energy available from the other sources, in this case, the body degrades proteins for energy. Please rewrite.
Lines 526-548: Please shorten this considerably (adhering to the principle »no results – no discussion«). In your study, you did not consider (why?) detrmining intramuscular fat content, although it is among the most important meat quality traits, also enabling to further explain many other traits. You can therefore not adhere to IMF for discussion on colour. Value b* (yellowness) may also be associated to oxidation, which in turn can be a consequence of unsaturated fat, originating from sources like soybean oil, which was used in the experiment.
Comments on the Quality of English LanguagePlease see comments to authors.
Author Response
- Language should be revised for style and grammatical corrections by an language expert. Sentences in some parts ar somewhat difficult to follow. The main issue of the article is the use and the interpretation of the words »low ambient temperature«. Although the pigs were reared in low temperature environment, the onset of the trial is not such, that would allow this comparison (i.e. all animals were reared at the same temeprature). The authors should therefore reconsider and thoroughly rewrite the manuscript (especially discussion) to avoid any conclusions suggesting the effects of low temperature. The introduction is rather lenghty, unconnected and unfocused. It should be concentrated solely on providing main points to justify the aim of the study. On the other hand, it is lacking essential information theSongilao pig breed (main characteristics, main environment to which the breed is adapted…). Same goes for the discussio, authors should improve focus, avoiding lenghty descriptions that are not directly linked to the results.
Response:Thank you for your suggestion, we have revised the relevant content.
- 3.Title: you mean low ambient temperature environment?
Response:Thank you for your suggestion, we have revised the relevant content.
- 4.Line 23: Please include the age.
Response:We are very grateful to Reviewer for reviewing the paper so carefully. We have revised the relevant content. Please check in lines 23.
- 5.Line 25: Please define the abbreviations CON and TES.
Response:Thank you for your suggestion, we have revised the relevant content. Please check in lines 25.
- 6.Line 31: were all animals slaughtered at the same time? What was their age and the duration of the trial?
Response:Thank you for your valuable feedback, we have revised the relevant content. Please check in lines 28-32.
- 7. Line 46: Please change the conclusions in line with the general comment.
Response:Thank you for your suggestion, we have revised the relevant content. Please check in lines 47- 50.
- 8. Line 119: What was the age and sex of the pigs included?
Response:Thank you for your valuable feedback, we have revised the relevant content. Please check them at line 109.
- Lines 122-131: Please include for all stages the time points or duraitons of the seperate phases . Were the changes in feed done at the same time points. Please also rewrite this part for better clarity.
Response:Thank you for your suggestion, we have revised the relevant content. Please check them at line 114-125.
- Line 123: what was the sex of the animals selected (castrate, gilt?). It should always be the same, not to interfere with the metabolic results.
Response:Thank you for your suggestion, we have revised the relevant content. Please check them at line 114-115.
- Line 128: It is better to adhere to the results of chemical composition (i.e. fat content) rather than confuse it with how much fat was added to the feed.
Response:Thank you for your suggestion, we have revised the relevant content. Please check them at line 123.
- Line 164: Please describe how was the feed intake recorded, was the feeding on ad lib basis? How did you ensure the ad lib allowance if the feeding was done in the limited periods (i.e. twice a day)?
Response:Thank you for your valuable feedback. Feed intake was recorded using the weigh-back method. Pigs were provided ad libitum access to feed via an automated feeding system. To ensure continuous feed availability, the feeders were replenished twice daily (8:00 AM and 4:00 PM) with fresh diets.
- Line 249: From vhich vein exactly?
Response:Thank you for your valuable feedback, we have revised the relevant content. Please check them at line 244-245.
- Line 251: What kind of tubes (anticoagulant) were used? If you are measuring glucose in blood, this additionally requires a glycolysis blocator.
Response:Thank you for your suggestion, we have revised the relevant content. Please check them at line 246. Glycolysis inhibitors were not employed in this study, as blood samples were processed and transported to the laboratory within a short timeframe, and all samples were maintained in the low ambient temperature during transport to suppress erythrocyte glycolysis.
- Lines 255-256: What is the reason for applying these tests?
Response:Thank you for your valuable feedback. In this experiment, the purposes of detecting alanine aminotransferase (ALT), aspartate aminotransferase (AST), total protein (TP), blood urea nitrogen(BUN), blood glucose (GLUC), high-density lipoprotein (HDL), low-density lipoprotein (LDL), and triglycerides (TG) are as follows:
- Liver Function Assessment (ALT & AST)
ALT and AST are sensitive markers of hepatocyte injury. Elevated activity indicates damage to the integrity of liver cell membranes (such as inflammation, toxicity, or metabolic abnormalities).
Relevance to Research Objectives: The feed components in this experiment (e.g., high fat supplementation) may stress liver metabolism. Monitoring ALT/AST can assess liver health, verifying feed safety.
- Protein Metabolism and Nutritional Status (TP & BUN )
TP and BUN reflects the synthetic capacity of the liver and the protein metabolism of the animal.
Relevance to Research Objectives: TP levels can indirectly assess feed protein utilization and liver synthetic capacity, providing a basis for optimizing feed formulations.
- Energy Metabolism and Glycemic Homeostasis (GLUC)
Blood glucose (GLUC) is a core indicator of energy metabolism, regulated by insulin, feed intake, and gluconeogenesis.
Relevance to Research Objectives: High-fat feed may alter blood glucose levels by affecting insulin sensitivity or gluconeogenesis pathways. Monitoring GLUC helps elucidate the adaptability of energy metabolism.
- Lipid Metabolism and Cardiovascular Health (HDL, LDL & TG)
HDL (high-density lipoprotein) and LDL (low-density lipoprotein) respectively reflect cholesterol reverse transport capacity and atherosclerotic risk, while TG (triglycerides) is a key indicator of lipid storage and mobilization.
Relevance to Research Objectives: The type of fat in the feed (e.g., soybean oil rich in unsaturated fatty acids) may influence the blood lipid profile. Monitoring HDL/LDL/TG can evaluate lipid metabolism efficiency and cardiovascular health risks. Combined with slaughter performance data (such as backfat thickness), a dose-response relationship can be established between dietary fat levels and lipid metabolism.
- Line 259: what was the lenght of the trial?
Response:Thank you for your valuable feedback, we have revised the relevant content. Please check them at line 255.
- Line 263:If the formula is for lean pigs, it does usually not cover the local breeds (fatty phenotypes). Please explain this (also in the discussion).
Response:Thank you for your suggestion, we have revised the relevant content. Please check them at line 520-522.
- Lines 273-279: Please provide methodology more in detail (i.e. reference, basic principles, number of technical repetitions, sampling location, sampling time, with colour measurement time of exposure to air is also important).
Response:Thank you for your suggestion, we have revised the relevant content. Please check them at line 274-301.
- Line 281:Please describe the statistical model and post-hoc tests.
Response:Thank you for your suggestion. When the experiment only includes a control group (CON) and a test group (TES), post-hoc tests are generally not applicable for directly comparing the differences between the two groups, because:
Purpose of Post-hoc Tests: In ANOVA or similar multi-group comparisons, post-hoc tests are used to determine which specific groups show differences after finding a significant difference overall (e.g., Tukey HSD is used for comparisons involving more than three groups).
Two-Group Scenario: The statistical methods for comparing two groups (such as t-tests) inherently test for differences directly, so there is no need for additional post-hoc analysis. Please check them at line 304.
- Table 7: Effects on slaughter performance
Response:Thank you for your suggestion, we have revised the relevant content. Please check them at line 374.
- Line 356 and Table 8 title: »composition consumes digestible protein…«? Rewrite.
Response:Thank you for your valuable feedback, we have revised the relevant content. Please check them at line 378 and 386.
- Line 385: What is the real implication of this part? Please shorten considerably.
Response:Thank you for your valuable feedback, we have revised the relevant content. Please check them at line 405 and 408.
- Line 395: By increasing the fat content, you also increase the energy content, which would block intake by itself in the long term. In addition dietary fats are the most potent regulator (blocator) of the apetite/feed intake. Please also consider that in the discussion, also in association to basic energy needs determined for the pigs from the experiment.
Response:Thank you for your suggestion, we have revised the relevant content. Please check them at line 409 and 413.
- Line 426: You are refering to chicken. Please try finding relevant literature specifically for pigs.
Response:Thank you for your suggestion, we have revised the relevant content. Please check them at line 434.
- Line 464: you are not showing any correlation in your results. Please rewrite.
Response:Thank you for your suggestion, we have deleted the relevant content. Please check them at line 470.
- Line 467: there are always some proteins degraded (obligatory protein loss) due to metabolism, turnover etc, even if there is enough energy available. The amount of protein that you are refering to would be replaced by fats if there was not enough energy available from the other sources, in this case, the body degrades proteins for energy. Please rewrite.
Response:Thank you for your suggestion, we have revised the relevant content. Please check them at line 476-483.
- Lines 526-548: Please shorten this considerably (adhering to the principle »no results – no discussion«). In your study, you did not consider (why?) detrmining intramuscular fat content, although it is among the most important meat quality traits, also enabling to further explain many other traits. You can therefore not adhere to IMF for discussion on colour. Value b* (yellowness) may also be associated to oxidation, which in turn can be a consequence of unsaturated fat, originating from sources like soybean oil, which was used in the experiment.
Response:Thank you for your suggestion, we have revised the relevant content. Please check them at line 538-547.
Reviewer 2 Report
Comments and Suggestions for Authors
This paper investigates the effects of optimising dietary energy supply for Songliao Black growing and fattening pigs in low ambient temperature environments.
The investigation's primary merit lies in its recognition of the significance of mitigating CO2 emissions in pig husbandry and the potential role of dietary alterations in realising this environmental objective.
Increasing fat and energy levels in pig diets can improve production performance and energy utilisation efficiency in cold environments.
The experimental group (TES) fed a high-energy diet exhibited lower feed intake but improved feed-to-gain ratio compared to the control group (CON), particularly during the fattening period. The TES group demonstrated higher crude fat digestibility and improved nitrogen utilisation, with lower urinary nitrogen content indicating reduced protein oxidation.
Pigs in the TES group exhibited enhanced energy deposition, particularly fat deposition, and reduced carbohydrate and protein oxidation. This suggests more efficient energy utilisation and reduced CO2 emissions (this is the most important result!).
The high-energy diet resulted in increased backfat thickness and improvements in certain meat quality parameters, such as higher yellowness values and lower shear force in the longissimus dorsi muscle.
In conclusion, the study demonstrates that elevating dietary fat and energy levels in cold environments can enhance production performance, energy and protein utilisation efficiency, and meat quality of Songliao Black pigs while potentially mitigating environmental impact through reduced CO2 emissions.
My questions are:
1. What were the specific differences in diet composition between the control and experimental groups?
2. Were there any negative effects or trade-offs observed with the high-energy diet in the experimental group?
Author Response
1. What were the specific differences in diet composition between the control and experimental groups?
Response:Thank you for your valuable feedback. The dietary differences between the two groups had been explained in the nutritional composition table.
2. Were there any negative effects or trade-offs observed with the high-energy diet in the experimental group?
Response:Thank you for your valuable feedback. No negative effects of a high-energy diet were observed during the experiment.
Round 2
Reviewer 1 Report
Comments and Suggestions for Authors
The authors have answered to most of the issues pointed out in the first manuscript review. Several of them, however, still remains to be answered. Please see the points below:
Line 62: write lower case letter in »the second«
Line 159:8: in your answer you mention that the feeding was done authomatically by a device (add also producer, city, county data), but you still mention in the text that the feeding was manual. Please rewrite providing more exact informaiton on the procedure.
Line 245: What do you mean by »precava«? Maybe vena cava cranialis? For taking blood this way, the pigs have to be restrained. Please improve this part of the description with this kind of information.
Line 286: in »pH«, the »p« is always lowercase and »H« always uppercase letter. Please rewrite.
Lines 274-301: Please add more description (basic principles) of the methods used. Please also provide proper references (cited on the end of the article).
Comments on the Quality of English LanguageThe manuscript still contains several grammatical and typing errors. Please read carefully and correct them.
Author Response
-
- Line 62: write lower case letter in »the second«
Response:We are very grateful to Reviewer for reviewing the paper so carefully. We have revised the relevant content. Please check in lines 62.
- Line 159:8: in your answer you mention that the feeding was done authomatically by a device (add also producer, city, county data), but you still mention in the text that the feeding was manual. Please rewrite providing more exact informaiton on the procedure.
Response:Thank you for your suggestion, we have revised the relevant content. Please check in lines 159-160.
- Line 245: What do you mean by »precava«? Maybe vena cava cranialis? For taking blood this way, the pigs have to be restrained. Please improve this part of the description with this kind of information.
Response:Thank you for your suggestion, we have revised the relevant content. Please check in lines 245-246.
- Line 286: in »pH«, the »p« is always lowercase and »H« always uppercase letter. Please rewrite.
Response:We are very grateful to Reviewer for reviewing the paper so carefully. We have revised the relevant content. Please check in lines 286.
- Lines 274-301: Please add more description (basic principles) of the methods used. Please also provide proper references (cited on the end of the article).
Response:Thank you for your suggestion, we have revised the relevant content. Please check in lines 271-313.
- The manuscript still contains several grammatical and typing errors. Please read carefully and correct them.
Response:Thank you for your valuable feedback, we have read carefully and corrected them.